# The Influence of Ambient Temperature on Adipose Tissue Homeostasis, Metabolic Diseases and Cancers

**DOI:** 10.3390/cells12060881

**Published:** 2023-03-12

**Authors:** Rehna Paula Ginting, Ji-Min Lee, Min-Woo Lee

**Affiliations:** 1Department of Integrated Biomedical Science, Soonchunhyang University, Cheonan 31151, Republic of Korea; 2Soonchunhyang Institute of Medi-Bio Science (SIMS), Soonchunhyang University, Cheonan 31151, Republic of Korea

**Keywords:** housing temperature, thermogenesis, brown adipose tissue, adipose tissue remodeling, obesity, metabolic disease

## Abstract

Adipose tissue is a recognized energy storage organ during excessive energy intake and an endocrine and thermoregulator, which interacts with other tissues to regulate systemic metabolism. Adipose tissue dysfunction is observed in most obese mouse models and humans. However, most studies using mouse models were conducted at room temperature (RT), where mice were chronically exposed to mild cold. In this condition, energy use is prioritized for thermogenesis to maintain body temperature in mice. It also leads to the activation of the sympathetic nervous system, followed by the activation of β-adrenergic signaling. As humans live primarily in their thermoneutral (TN) zone, RT housing for mice limits the interpretation of disease studies from mouse models to humans. Therefore, housing mice in their TN zone (~28–30 °C) can be considered to mimic humans physiologically. However, factors such as temperature ranges and TN pre-acclimatization periods should be examined to obtain reliable results. In this review, we discuss how adipose tissue responds to housing temperature and the outcomes of the TN zone in metabolic disease studies. This review highlights the critical role of TN housing in mouse models for studying adipose tissue function and human metabolic diseases.

## 1. Introduction

During the past decade, adipose tissue has attracted the attention of metabolic researchers as an endocrine and thermoregulator tissue. Beyond its general function in energy storage, the endocrine and thermoregulatory functions of adipose tissue explain the more significant roles of adipose tissue in regulating systemic metabolism [1]. Adipose tissue consists of several cell types, including adipocytes, preadipocytes, immune cells and endothelial cells, which produce various bioactive compounds called adipokines [2]. Changes in the environment, including metabolic status, are known to alter adipose tissue at both the physiological and molecular levels to adjust to new tissue homeostasis. For example, high-fat diet (HFD)-induced obesity is known to change the adipose tissue cell population and induce inflammatory gene expression in most adipose cell types, leading to systemic inflammation and insulin resistance [3,4]. Other conditions, such as a cold environment or the activation of β-adrenergic receptors (βARs), lead to the activation of brown adipose tissue (BAT) and beige adipose tissue (BeAT) thermogenesis, which correlates with the metabolic improvement in both mice and human [5,6,7].

The study of the role of adipose tissue in maintaining energy balance is of great importance for the treatment of obesity and its related metabolic diseases. However, most metabolic disease studies using mice were conducted under ambient temperatures around 22–24 °C, referred to as room temperature (RT), where the mice are chronically exposed to cold stress. RT housing potently activates the sympathetic nervous system (SNS) to trigger β-adrenergic signaling in rodents [8,9]. Thermoneutrality is defined as the temperature range in which the body has minimal thermogenic activity and maintains its metabolic rate [10,11]. It is known that mice housed in their TN zone have a lower metabolic rate compared to the mice housed at RT [12]. Animal studies have confirmed that housing temperature dramatically affects the function of adipose tissue and other metabolic organs, as well as the pathophysiology of diseases. Therefore, it is still questionable whether mice living under constant cold stress are translationally relevant models that mimic human metabolic diseases living primarily in thermoneutral (TN) conditions (approximately 30 °C for mice).

Recently, several studies have attempted to elucidate the consequences of TN housing in mice physiology and pathology. Since housing temperature cannot be argued to be an optimal condition for studying human disease in mouse models, it can be considered one of the important factors influencing the results and data interpretation when modeling diseases in mice. In this review, we discuss the effect of housing temperature on adipose tissue function and its consequences on systemic metabolism. We focus on the importance of housing temperatures in preclinical metabolic disease research, as well as significant differences in the phenotypes and pathology in mouse models in response to ambient temperatures. This review highlights the critical role of ambient temperature in adipose tissue metabolism; thus, it is important to physiologically mimic human conditions in metabolic studies using mouse models.

## 2. Ambient Temperature on Mouse Metabolism

Mammals have several heat-gain mechanisms to maintain their core body temperature in response to a thermal challenge. When immediate reactions (e.g., cuddling, piloerection, or vasoconstriction) are insufficient during hypothermic conditions, thermogenic action is required to maintain a stable core temperature [13]. Thermogenesis has two categories: shivering and non-shivering thermogenesis (NST). Shivering thermogenesis (ST) acts as an acute response to thermal stress. It requires the continual contraction-relaxation of muscle, which leads to muscle injury and limits the animal’s movement during prolonged cold [14,15]. NST occurs mainly in BAT and is preferably used as a long-term strategy during cold challenges [16]. When NST alone is adequate to generate heat, such as during chronic mild cold exposure, shivering becomes unnecessary or minimal [17].

The difference in metabolic rates observed in different housing temperatures is the consequence of mice’s thermogenic response to a thermal challenge. The gradual decrease in mice housing temperature from 30 °C (TN) to ~20–16 °C (mild cold) and ~5 °C (severe cold) leads to a gradual increase in mice oxygen consumption [18,19]. When the environment temperature is below the TN zone, the metabolic rate increases as energy is required to maintain body temperature. As the ambient temperature exceeds the TN zone, the energy expenditure (EE) is enhanced owing to the body cooling mechanisms [20,21,22]. The correlation between the metabolic rate and ambient temperature is illustrated in Figure 1. Therefore, the application of TN conditions in pre-clinical metabolic studies is growing as researchers become aware of the importance of housing temperature in creating a physiologically relevant metabolic model for humans.

Several studies have compared the physiological and metabolic differences of mice housed at RT and near their TN zone. Normal wild-type mice housed in the TN zone had lower metabolic rates and consumed less food but maintained similar body weight to the mice housed in RT. Despite a lower metabolic rate in TN housing, diet-induced obesity (DIO) mice did not have lower food intake and had higher body weight than mice housed at RT after 33 days of experiment [18]. Another study demonstrated that the activation of β-adrenergic signaling by reducing the ambient temperature from 30 °C to 22 °C for approximately three days led to higher adipose tissue thermogenic activity, resulting in significant weight loss in *ob/ob* mice [9].

The application of TN housing in animal models is still debatable, as there is no standardized temperature to define the TN zone for mice due to different mouse phenotypes and environmental conditions. Stemmer et al. demonstrated that nude mice on a C57BL/6 background had higher EE than wild-type mice when housed at 23 °C, whereas 33 °C housing abolished the EE difference. The hairless phenotype in nude mice reduces heat insulation. Therefore, to maintain their body temperature, nude mice may activate the compensatory thermogenic program, such as muscle-mediated ST and BAT and BeAT-mediated NST, which lead to higher EE [23]. Other factors, such as the number of animals per cage, age and bedding, are crucial in defining ambient temperature [12].

A recent study suggested that strictly studying mice at thermoneutrality was nearly impossible. They found that the mouse TN point was 29 °C in the light phase and 33 °C in the dark stage, with a diurnal change of 4 °C [24]. They recommended using 28–29 °C to model TN, as mice can self-regulate EE and body temperature while minimizing cold-induced thermogenesis [24]. Others suggested using 25–27 °C to study mice whose EE is 1.7× basal metabolic rate (BMR), which is similar to typical human EE [25,26,27]. Altogether, these results indicate that ambient temperature plays a critical role in defining the metabolic phenotypes of mice.

## 3. Adipose Tissue Thermogenic Function in Mice and Human

Based on their functions and morphology, adipose tissue is categorized into white adipose tissue (WAT) and BAT. WAT consists mostly of white adipocytes with one large lipid droplet or unilocular cells. WAT will store excess energy in the form of triglycerides (TG) which will be broken down into fatty acids and glycerol to provide energy to other organs when needed [28]. Oppositely, BAT dissipates energy to generate heat through βARs activation or cold exposure. Classic BAT consists of multilocular adipocytes with small lipid droplets and abundant mitochondria consisting of high expression of uncoupling protein 1 (UCP1) constantly [29].

In mammals, BAT is mainly observed in neonates and plays an important role in maintaining body temperature along with behavior thermoregulation, such as cuddling [30,31]. BAT activation improves obesity-induced metabolic impairment, as observed in hig mice transplanted with BAT. Conversely, a lack of brown adipocytes leads to an increase in body weight, adipose tissue inflammation and insulin resistance [32]. The appearance of BAT in adult humans is mostly observed under cold environments or βARs stimulation [5,6,33]. In adult humans, BAT can be found in the upper trunk region (i.e., interscapular, supraclavicular, paravertebral and pericardial) and, to some extent, mediastinal and mesenteric areas [34,35]. Yoneshiro et al. demonstrated that a 2-h cold exposure at 17 °C to healthy young male subjects for six weeks increased BAT activity and EE, which was negatively correlated with body fat mass [5]. Chronic mirabegron treatment, a β3-adrenergic receptor agonist, to healthy young females induced BAT metabolic activity without changes in body weight. Mirabegron-treated subjects also improved insulin sensitivity and insulin secretion [6]. During cold exposure, BAT glucose uptake rate was ~3 µmol/min in healthy subjects and reduced up to ~0.1 µmol/min in obese subjects with or without type 2 diabetes [36].

Although BAT is a major and distinct form of thermogenic adipose tissue, thermogenesis is not restricted to this adipose tissue. Inducible BAT, commonly known as BeAT, is found within inguinal white adipose tissue (iWAT). Unlike BAT, BeAT thermogenic function is activated only under βARs stimulation or cold exposure [37]. In the in vivo models, recruitment of BeAT in iWAT improved obese conditions by lower body weight gain [7]. The appearance of BeAT has been observed in subcutaneous WAT (scWAT) depots of humans. Cold or mirabegron treatment induced UCP1 protein in abdominal scWAT of lean and obese subjects. Other than cold and βARs activation, BAT or BeAT activity can be induced by interleukin (IL)-6 [38], PPARg ligands (such as PAF-metabolites) [39,40,41], mitochondrial RNA [42] and also the energy oversupply from high-calorie or HFD [43].

Besides WAT, BAT and BeAT, several adipose tissue depots were identified; perivascular adipose tissue (PVAT), epicardial adipose tissue (EAT) and perirenal adipose tissue (PRAT) were involved in the pathogenesis of diseases, such as hypertension [44], atherosclerosis [45], renal and cardiovascular system [46]. Moreover, human PRAT contains brown adipocyte precursor cells [47]. In humans, PRAT is visceral WAT (vWAT) similar to iWAT [48], and cold weather induced the browning of PRAT [49]. BAT-specific genes (*Prdm16*, *Ucp1* and *Cpt1b*) are expressed in PVAT from human coronary arteries to the degree that appeared in the iWAT and BAT of mice. These observations suggest that PVAT has a similar characteristic to BeAT [50]. In rodents, PVAT from the abdominal aorta displays characteristics of WAT, whereas PVAT surrounding the thoracic aorta is known to express some BAT-specific genes [51,52]. However, future study is essential to clarify that these distinguishable adipose tissues contribute to maintaining body temperature and EE compared with BAT and BeAT.

## 4. BAT Whitening upon Thermoneutral Housing

Typical BAT from mice housed at RT has small multilocular brown adipocytes that are evenly spread out in the tissue, whereas BAT from mice housed in the TN zone consists of unilocular brown adipocytes and has a similar appearance to white adipose tissue (WAT) in mice housed at RT, which is called whitening of BAT. In addition to BAT, iWAT, the central depot for beige adipocytes, undergoes morphological changes at different ambient temperatures. In cold environments, iWAT primarily consists of multilocular adipocytes, indicating a browning event. The browning in iWAT completely disappeared upon TN housing, as confirmed by the appearance of unilocular adipocytes, which were larger than those in mice housed at RT. Following morphological transformation, the expression of brown adipocyte-specific genes, such as *Ucp1*, *Pgc1a* and *Cidea*, was decreased in TN compared to RT, even though it was still higher than that in iWAT [53].

Furthermore, a reduction in mitochondrial density and mitochondrial proteins is observed in whitened BAT [54]. Mitochondrial mass is tightly controlled by both mitochondrial biogenesis and autophagy-mediated mitochondrial degradation [55,56]. In the TN zone, BAT prefers to reduce mitochondrial mass via the reduction of mitochondrial biogenesis-related gene expression and the alteration of lysosome- and autophagy-related gene expression. Transcription factor EB (TFEB), a transcription factor that regulates lysosomes and autophagy, plays a critical role in BAT whitening in TN housing [54,57,58]. Brown adipocyte-specific TFEB deficiency attenuated TN-induced BAT whitening, as confirmed by the smaller lipid droplets and higher UCP1 and oxidative phosphorylation (OXPHOS) complex protein levels. However, loss of TFEB in brown adipocytes did not alter the response of mice to cold, nor did it protect mice from DIO [54]. This autophagy-mediated mitochondrial clearance was also observed during the beige-to-white adipocyte transition. Inhibiting autophagy by deletion of *Atg5* or *Atg12,* specifically in UCP1^+^ adipocytes, was capable of maintaining beige adipocyte thermogenic function even after withdrawal of external stimuli (e.g., β3AR agonist) and protected the mice from DIO [59].

Another hallmark of BAT whitening is the accumulation of TG [53] via several metabolic pathways. Schlein et al. recently demonstrated that lipid accumulation in BAT whitening mainly originated from newly synthesized fatty acids rather than dietary fatty acids. They found that mice with depleted lipogenic transcription factor carbohydrate-response element binding protein (ChREBP) were resistant to lipid accumulation and mitochondrial breakdown upon TN-induced BAT whitening. They also found that autophagy-mediated mitochondrial breakdown in BAT whitening was ChREBP-dependent, as wild-type mice had a higher number of mature autophagosomes than ChREBP-knockout (KO) mice. However, excess dietary lipids diminished the ChREBP deficiency-induced protective effects in brown adipocytes regarding lipid accumulation [60]. These studies demonstrated dynamic changes of BAT in response to different ambient temperatures (Figure 2).

A significant question is whether BAT from mice housed in their TN zone is physiologically relevant to human BAT and can be used to study human metabolic diseases. De Jong et al. use the term “physiologically humanized” mice for mice housed in their TN zone. They found that BAT from mice housed in their TN zone was morphologically similar to human supraclavicular BAT, reflecting the existence of unilocular adipocytes surrounded by multilocular adipocytes [53]. However, the UCP1 protein was still detectable in TN BAT, indicating that the cellular identity of brown adipocytes was preserved under this condition [60]. Other than TN housing, BAT whitening can be induced by multiple factors, such as ambient temperature, βARs signaling impairment, leptin receptor deficiency, lipase deficiency and genetic modification (e.g., *ob/ob* mice) [61,62]. Therefore, it is still debatable whether “physiologically humanized” is a proper term to define BAT morphology in TN housing. Instead, the BAT transformation in TN housing is more closely related to changes in the functionality of brown adipocytes than that of cellular identity. Roh et al. previously demonstrated that warming only modestly changed brown adipocyte transcriptomics and maintained a static chromatin state, although BAT showed a whitening phenotype [63]. Therefore, further investigations on the lineage of brown adipocytes using human samples can help answer the significant question asked above.

## 5. Adipose Tissue and Systemic Inflammation

Adipose tissue is a heterogeneous tissue consisting of a mature adipocyte fraction and a stromal vascular fraction (SVF), which contains various cell types, including adipocyte precursor cells (APCs), fibroblasts and immune cells. The profile of adipose tissue-resident cells, especially immune cells, is dynamically altered by environmental changes, which contributes to the overall function and remodeling of adipose tissue.

Exposure to cold temperatures induces the alternative polarization of macrophages in BAT and BeAT. It is suggested that these alternatively activated macrophages contribute to thermogenesis by producing catecholamines which induce thermogenesis by directly activating β3-adrenergic signaling in adipocytes [64]. In addition, Pirzgalska et al. demonstrated that the subpopulation of macrophages named sympathetic neuron-associated macrophages (SAMs) also regulate catecholamines contents in adipose tissues. These macrophages directly metabolize catecholamines via the norepinephrine (NE) transporter solute carrier family 6 member 2 (SLC6A2) and the NE degradation enzyme monoamine oxidase A (MAOA) [65]. These macrophages may contribute to the local titers of NE produced by the sympathetic neurons in adipose tissue.

In obesity, necrosis-like adipocytes trigger an inflammatory response, resulting in the massive recruitment of monocytes, which polarize into pro-inflammatory M1 macrophages and form crown-like structures (CLS) [66]. These M1-polarized macrophages secrete several inflammatory cytokines, such as monocyte chemoattractant protein-1 (MCP-1), tumor necrosis factor-α (TNF-α), IL-1β, IL-6, and chemokines, contributing to the recruitment of incoming monocytes [67,68]. Through single-cell RNA sequencing (scRNASeq) analysis, Jaitin et al. discovered an obese-specific macrophage subpopulation that enriched lipid metabolisms and phagocytosis-related genes. This macrophage subpopulation was marked by the high expression of *Cd9* and *Trem2* [69]. An additional study found that *Cd9* and *Trem2*-positive macrophages are accumulated in obese human WAT [70]. Compared to WAT, little is known about the function of immune cells in BAT during obesity. HFD-induced macrophage infiltration was also observed in BAT, although at a lower degree than WAT [71]. These infiltrated macrophages are characterized as pro-inflammatory M1 macrophages and are capable of suppressing the induction of UCP1 in BAT and iWAT [72,73].

Even though immune cell plasticity has been well characterized in adipose tissues, the contribution of TN housing in regulating immune cell composition is limited. Two independent research groups have observed a higher MAC-2 (galectin-3) positive macrophage accumulation in BAT of C57BL/6 lean mice housed at TN compared to the RT housing or cold conditions. These macrophages were mostly aggregated into CLS (Figure 2) [61,74]. Similar phenotypes were also observed in leptin receptor-deficient, β-less and *Atgl^−/−^* mice [61]. However, cold acclimatization reversed TN-induced macrophage accumulation in BAT. Interestingly, after prolonged TN housing, macrophage infiltration in BAT did not interfere with BAT thermogenic function. The expression of genes related to thermogenesis in response to cold acclimatization was imperceptible between prolonged TN housing and standard RT housing in mice [74]. These results indicate that the accumulation of macrophage in BAT is not correlated with BAT competence.

Alteration of immune cell composition in TN housing also occurs systemically. For example, LyG6^+^ monocytes accumulate in the bone marrow but decrease in circulating blood in TN-housed mice compared to the RT-housed mice [75]. TN housing exacerbated TNF-α and IL-6 levels in the serum of mice at baseline and LPS-stimulated condition [76]. Furthermore, TN housing accelerates the infiltration of immune cells into the tumor microenvironment (TME) [77]. The relationship between ambient temperature and immune responses can be interpreted as a concept of priority. In RT housing, where the mice are chronically exposed to mild cold, they prioritize energy resources to produce heat rather than immune responses [78,79]. Cold exposure results in fewer activated monocytes, which leads to the suppression of pathogenic T-cell priming and the reduction of T-cell cytokine expression in autoimmunity [80,81]. These results indicate that mice housed at RT and TN have a significant difference in the basal systemic immune response.

## 6. Ambient Temperature in Metabolic Disease Models

As discussed before, TN housing affects mice’s physiological and metabolism, which impacts the development of the diseases observed in mice (Table 1). In this section, we provide information comparing the disease development and phenotypes observed between mice housed in RT and their TN zone.

### 6.1. Obesity and Insulin Resistance

It is well described that obesity-induced adipose tissue and systemic inflammation are associated with insulin resistance [88]. Previous studies suggested that inflammation is initiated in adipose tissue and then spreads to other peripheral tissues, resulting in systemic insulin resistance [89,90]. Moreover, obesity interferes with adipose tissue’s capacity to expand properly and leads to ectopic fat deposition in insulin-sensitive tissues such as the liver, heart and skeletal muscle. The accumulation of lipids in these tissues affects lipid uptake and metabolism, thereby contributing to the development of insulin resistance [91,92].

As mentioned previously, mouse metabolism is different between the RT and TN housing. However, the outcomes of TN housing using the DIO model remain unclear. Stemmer et al. showed that TN housing promoted obesity in C57BL/6 nude mice, whereas it was resistant to obesity in RT housing. Higher adiposity, increased hepatic fat accumulation, lower glucose clearance and worsened glucose intolerance were observed in HFD-fed C57BL/6 nude mice housed in the TN zone but not in RT [23]. Tian et al. demonstrated that TN housing accelerated immune cell infiltration as well as M1 and M2 macrophage accumulation in epididymal WAT (eWAT) as early as three weeks after HFD exposure. Whereas significant accumulation of immune cells in eWAT was observed after 12 weeks after HFD initiation in mice housed at RT [82]. Similar phenotypes were also observed in BAT, where TN housing accelerated immune cell accumulation in the thermogenic adipose tissue [82]. Interestingly, despite earlier immune cell infiltration, there were no differences in glucose and insulin tolerance between mice housed at TN and RT, both in acute (3–4 weeks) and chronic (over 10 weeks) HFD exposure [82]. These findings suggest that TN housing uncouples inflammation from obesity-induced insulin resistance. Another study by Clayton and McCurdy demonstrated that short-term TN housing combined with an acute HFD (1 day) induced body weight and adiposity. Prolonged HFD exposure (5 days) ameliorated the body weight difference between RT and TN housing. They also examined the effects of temperature on BAT and soleus muscle glucose metabolism. They found that glucose uptake during insulin-stimulated conditions was highest in the soleus muscle in a TN condition, whereas BAT had the most significant uptake at RT. One-day HFD was enough to reduce BAT glucose disposal up to 63% at 22 °C but not 30 °C. In contrast, soleus glucose disposal was reduced by 54% by HFD feeding at 30 °C [83].

The inconsistent results observed in these three studies may be explained by differences in mouse strains, temperatures used for RT and TN conditions and acclimatization period in TN housing, which are essential for obesity research [23,82,83]. The age difference is another contributor to controversies in animal feeding studies. Several studies have demonstrated that dietary exposure in young animals may induce less damage than that in adulthood [93,94,95].

The diet-dependent metabolic changes observed in mice housed in their TN zone were closely related to the inactivation of the NE-mediated BAT activation pathway. Similar phenotypes were also observed in mice housed at RT with genetic modification or exogenous inhibitor treatment [96,97,98,99,100]. BMP7 (an inducer of BAT differentiation) treatment in C57BL6 mice resulted in increased BAT weight, increased expression of *Ucp1*, higher EE and food intake, lower WAT weight and higher lipolysis at RT, but not in their TN zone [100]. Previous studies showed that mice with UCP1 depletion resisted DIO [96]. However, further experiments demonstrated that DIO resistance disappeared with TN housing because of the lower metabolic efficiency [97]. The lack of type 2 deiodinase enzymes required for tyrosine hydroxylase activity had a similar effect to UCP1-KO in mice. The DIO resistance phenotype was observed only in RT housing but is exacerbated in TN conditions [99].

### 6.2. Cardiovascular Physiology and Atherosclerosis

Cardiac output is the product of heart rate and stroke volume and is the mechanism by which blood flows around the body to provide the oxygen demands in every tissue [101]. The volume of blood pumped by the heart depends on metabolic needs. For example, oxygen demand increases during exercise to fulfill tissue oxygen requirements, whereas the opposite occurs during sleep when oxygen demand and heart rate drop [22]. As ambient temperature alters the metabolism in mice, it can also change cardiovascular physiology. The resting heart rate of mice is reduced from approximately 550–600 beats/min at 20–22 °C to approximately 300 beats/min at 30 °C [102,103,104]. This reduction may be due to the decreased thermogenesis demand at 30 °C. An arterial blood pressure change was also observed, which was 105 mmHg at 20 °C and reduced to 75 mmHg at 30 °C [102]. Therefore, the basal level of cardiovascular physiology is vastly changed from RT to TN housing, which should be considered in human disease studies. Several connections between the alteration of metabolic activity and the cardiovascular system can influence the development of cardiovascular diseases, such as atherosclerosis [105].

Atherosclerosis is a chronic disease in which arteries harden because of the accumulation of cholesterol-rich plaques in the subendothelial vessel walls. Diabetes, dyslipidemia, hypertension and cigarette smoking are common risk factors for atherosclerosis. High circulating plasma cholesterol levels lead to a change in arterial endothelial permeability, which enables the migration of low-density lipoprotein (LDL)-C particles into the arterial walls. These LDL-C particles are then oxidized to become chemoattractants [106]. In addition, circulating monocytes recognize and bind to adhesion molecules, such as vascular adhesion molecule-1 (VCAM-1) and selectins expressed by endothelial cells, thereby migrating into the subendothelial space. Thereafter, monocytes acquire macrophage characteristics and become foamy macrophages [106,107]. Therefore, atherosclerosis is also an inflammatory disease.

Several mechanisms link adipose tissue with the development of atherosclerosis in obesity. Adipokines and metabolites secreted by adipose tissue can alter the levels of inflammatory factors, clotting factors and circulating lipoproteins. These factors affect endothelial cells, arterial smooth muscle cells and macrophages/monocytes, which contribute to the atherogenic environment of the vessel wall [108]. In obesity, adipose tissue secretes a high level of MCP-1, an inflammatory mediator, contributing to increased circulating MCP-1. High circulating MCP-1 is known to increase the number of circulating CD11b monocytes, which are involved in the attachment of monocytes/macrophages to the vascular wall [109]. This condition exacerbates vascular inflammation and atherosclerosis. Obesity increases fatty acid release from adipose tissue, resulting in higher very-low-density lipoprotein, TG and apolipoprotein B (apoB) secretion [110,111,112]. Accumulation of free fatty acids in peripheral tissues can exacerbate insulin resistance and systemic inflammation through Toll-like receptor (TLR) 4, resulting in further development of atherosclerosis [113,114,115]. Activation of BAT enhances fatty acid uptake, resulting in improved hepatic clearance of cholesterol-enriched remnants [116]. This indicates that the ability of BAT to clear systemic fatty acids can ameliorate hyperlipidemia and protect against atherosclerosis.

TN conditions exacerbate metabolic inflammation in obesity; however, the effects on vascular inflammation and the pathophysiology of atherosclerosis remain unclear. Tian et al. demonstrated that TN housing enhanced the accumulation of atherogenic lesions in *Apoe^−/−^* mice fed a Western diet. They observed increased infiltration of macrophages and dendritic cells (DCs) into the vessel wall and PVAT [82]. Using atherosclerosis-resistant C57BL/6J mice, Giles et al. found evidence of small atherogenic lesions in the aortic root of mice housed at 30 °C, which was not observed in mice housed at 23 °C. They also found that mice housed in their TN zone had a substantial fraction of the total cholesterol in LDL, similar to that in humans [84]. These two studies found that TN housing enhanced the development of atherogenic plaques despite improving hemodynamic parameters such as heart rate and blood pressure.

### 6.3. Non-Alcoholic Fatty Liver Diseases (NAFLD)

NAFLD is a common chronic liver disease that leads to the development of hepatocellular carcinoma (HCC) and liver transplantation. NAFLD is defined as approximately 5–10% fat accumulation by liver weight in the absence of other causes of steatosis [117,118]. As discussed previously, the secretion of fatty acids from adipose tissue is increased in obesity, leading to ectopic fat deposition in the liver [91,92]. The accumulation of fatty acids in the liver reduces hepatic insulin clearance and induces gluconeogenesis and TG synthesis. The alteration in adipokine release by adipose tissue also contributes to the development of NAFLD [119]. In obesity, an increase in adipose tissue mass leads to higher secretion of leptin. Leptin can regulate hepatic fibrosis by inducing α2 collagen gene expression [120,121]. Higher production of pro-inflammatory cytokines, such as TNF-α and IL-6 in adipose tissue, contributes to the severe inflammatory environment of the liver [122].

Despite the severity of NAFLD, the molecular mechanisms underlying NAFLD pathogenesis are inconclusive because of the lack of animal models that can colligate the disease spectrum in humans [76,123]. Previously, female mice were considered resistant to NAFLD development due to hormonal differences. However, housing the mice under TN conditions alongside exposure to HFD resulted in higher pro-inflammatory cytokines, lower corticosterone production and exacerbated diet-induced NAFLD, independent of sex [76]. They demonstrated that TN housing increased intestinal permeability and altered the microbiome profile. TLR4 deletion, activation of the IL-17 axis and depletion of gram-negative bacteria ablated all TN-related phenotypes [76]. This study uncovered a correlation between ambient temperature and the composition of the gut microbiome in the pathophysiology of NAFLD. A higher composition of gram-negative bacteria in TN-housed mice increases intestinal permeability and leads to higher interaction between LPS and TLR4 to increase inflammation [76]. A previous study discussed how cold stress remodeled the small intestinal microbiome [124]. However, further studies are necessary to determine the effects of TN housing on the gut microbiome.

### 6.4. Cancer

The contribution of innate and adaptive immune cells in cancer studies has been widely explored and is known to play a critical role in tumor surveillance and antitumor immunity, as well as in supporting tumor growth [125,126,127]. Antitumor immunity is mainly mediated by CD8^+^ T helper cells and natural killer (NK) cells, whereas macrophages support tumor growth and metastasis [22]. Adipose tissue malfunction and chronic inflammation in obesity are the major causes of adiposity-induced tumorigenesis. Adipokines, such as leptin, IL-6 and TNF-α, have been reported to contribute to tumor development [128,129]. Leptin content is higher in patients with prostate cancer than in those with benign prostate hyperplasia, indicating that leptin functions as a biomarker for prostate cancer staging [130,131]. Proinflammatory cytokines released by adipose tissue also contribute to cancer development. Accumulation of CLS macrophages was observed in the adipose tissue of patients with gastrointestinal cancer, indicating a possible correlation between adipose tissue inflammation and cancer progression [132,133]. Activation of BAT during cold exposure increases glucose uptake into BAT, leading to a reduction in glucose availability for tumor disposal [134]. These studies demonstrate that crosstalk between adipose tissue and the tumor microenvironment contributes to cancer progression.

However, all studies demonstrating the relationship between adipose tissue inflammation, systemic inflammation and cancer progression were conducted in thermally stressed animal models, which might not represent the mechanisms in humans. TN housing reduced tumor growth and metastasis in mice inoculated with colon carcinoma, skin melanoma, pancreatic carcinoma, or mammary gland adenoma tumor cells compared to RT housing [77,85]. These phenotypes are caused by alterations in the number, activity and function of DCs and CD8+ T cells in the TN environment. Mice housed in their TN zone have more activated CD8+ T cells, and higher interferon-gamma (IFN-γ) expression, whereas mice housed at RT have more immature DCs [77,85]. Mice chronically exposed to cold (e.g., RT) have elevated NE levels and prioritize the energy demand for thermogenesis rather than the immune response to tumor cells [86,135,136]. Circulating NE levels are higher in mammary gland adenoma murine models housed under RT than that housed in TN. The release of NE-activated myeloid-derived suppressor cells (MDSCs) through βAR leads to a reduction in T-cell proliferation and exacerbates tumor growth. βAR KO ablates these phenotypes at RT or with the treatment of βAR agonists under the TN condition [137]. TN-induced adipose tissue inflammation may also contribute to systemic inflammation and affect the tumor microenvironment. Hormone and metabolic reprogramming in adipose tissue can affect tumor growth both positively and negatively. Chronic cold-induced fibroblast growth factor 21 (FGF21) and fatty acid metabolism can trigger tumor growth, whereas competition for glucose represses tumor development [79]. Therefore, TN housing is an important factor in defining the relationship between adipose tissue and cancer progression.

The TN condition improves irradiation and chemotherapy responses in mice with colon adenocarcinoma [86], indicating a critical role of ambient temperature in cancer therapy studies. In an irradiation-induced cancer model, hematopoietic stem cells tended to undergo apoptosis when irradiation was conducted at TN temperature compared to RT [87]. These studies suggest that TN housing might improve the sensitivity of mice to cancer therapies and improve our understanding of the pathophysiology of cancer.

## 7. Concluding Remarks

Adipose tissue is a dynamic tissue that is responsive to external stimuli to maintain homeostasis. Adipose tissue remodeling due to external stimuli can affect the homeostasis of other tissues and whole-body metabolism. As a metabolic organ, adipose tissue is essential in the pathophysiology of metabolic diseases, such as obesity and its related diseases. However, most disease studies using mouse models are conducted at RT, where mice are chronically exposed to mild cold. Therefore, our understanding of the pathophysiology and mechanisms of diseases likely came from studies using “cold-stressed” mice rather than mice in a “basal” state. Housing mice in RT increases their metabolic rate, which is equivalent to intense exercise in humans. As this condition creates a biological gap between mice and humans, the application of TN housing in mice models should be considered for studying human diseases.

In this review, we discuss the critical role of TN housing in the physiology and function of mouse adipose tissue. TN housing induces adipose tissue remodeling, characterized by a reduced mitochondrial number and higher accumulation of lipids in BAT. In addition, TN housing promotes immune cell infiltration into WAT and BAT in lean and obese mice. Not only adipose tissue inflammation, but TN housing also accelerates systemic inflammation, which leads to the different phenotypes observed in mice with diseases related to immune response, such as atherosclerosis and cancer. Moreover, TN conditions tend to improve the efficacy of chemotherapy and exert a beneficial effect on anti-tumorigenesis in several cancer models. Thus, further studies are required to examine the influence of TN housing on more diverse types and developmental stages of cancer.

This review highlights that ambient temperature critically contributes to the outcomes of murine studies, both in physiological and disease models (Figure 3). Despite various factors that affect TN housing for mice, such as the number of animals per cage, cuddling, inexact TN temperatures and period of pre-acclimatization, we conclude that studying the role of TN housing in mouse metabolic disease models is worth the effort to better understand human diseases.

## Figures and Tables

**Figure 1 cells-12-00881-f001:**
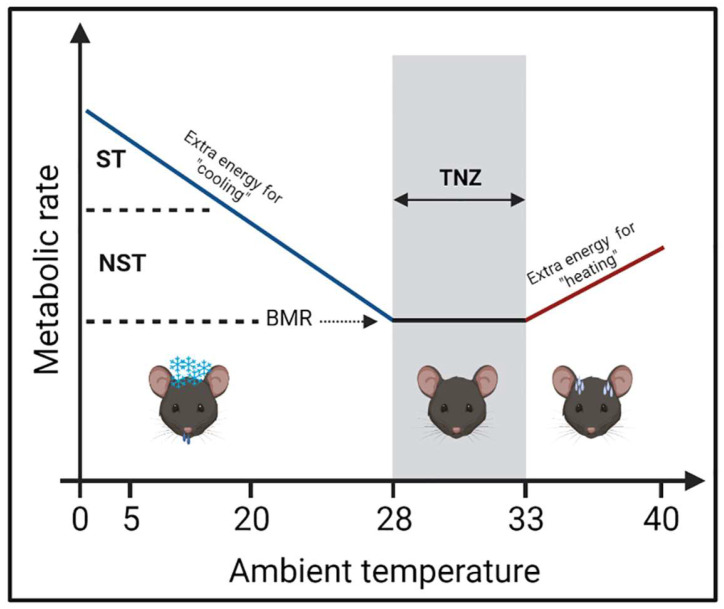
Thermoregulation of mice in response to ambient temperature. In ambient temperature below their TNZ, mice need extra energy for the “heating” process to maintain their body temperature. Oppositely, mice need extra energy for “cooling” when they are housed in an environment above their TNZ. In their TNZ, mice can maintain their basal metabolism without any requirement for thermogenic activity. BMR: basal metabolic rate; ST: shivering thermogenesis; NST: non-shivering thermogenesis; TNZ: thermoneutral zone.

**Figure 2 cells-12-00881-f002:**
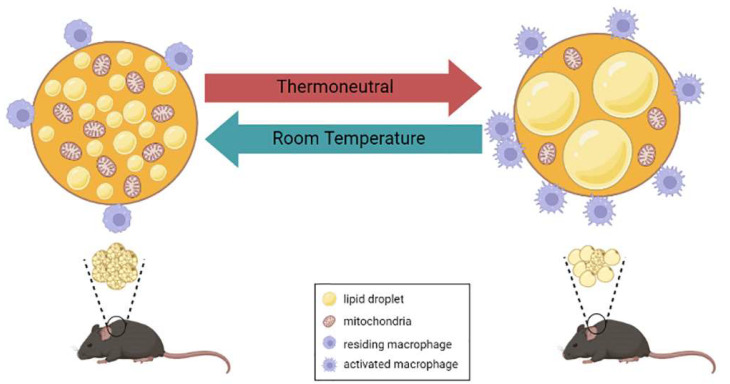
Phenotypes alteration of brown adipocytes in different housing temperatures. In standard animal housing (room temperature or ~21–24 °C), brown adipocytes contain multilocular lipid droplets with a high number of mitochondria. A small number of immune cells, such as macrophages, are known to reside in BAT. Housing the mice in the thermoneutrality (~28–31 °C) alters brown adipocytes phenotypes, observed by the appearance of unilocular lipid droplets and a low number of mitochondria. Higher infiltration of macrophages was also observed in BAT from mice living in the TN zone. These infiltrated macrophages reside in their activated state, reflected by the crown-like structure. Mechanisms such as autophagy-mediated mitochondria clearance and ChREBP-dependent lipid accumulation contribute to thermoneutrality-induced BAT remodeling.

**Figure 3 cells-12-00881-f003:**
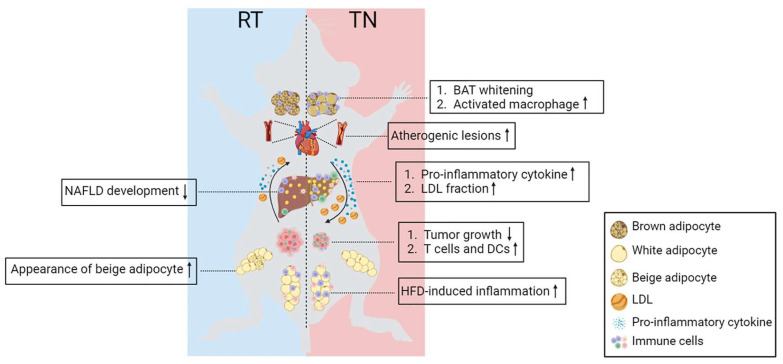
TN housing alters mice’s phenotype and response to the development of diseases. TN housing induces BAT and reduces the appearance of beige adipocytes in iWAT. Upon HFD feeding, TN housing exacerbates the accumulation of immune cells in eWAT and the liver. Housing the atherosclerosis mice model in their TN zone along with western-diet feeding leads to a greater accumulation of atherogenic lesions. TN housing reduced tumor growth in mice inoculated with tumor cells. These phenotypes might be due to the higher number of CD8+ T cells and activated DC infiltrated into the tumor cells. TN housing enhances circulating pro-inflammatory cytokines resulting in higher systemic inflammation. Furthermore, TN housing also alters the circulating cholesterol profile reflected by a higher fraction of LDL. RT: room temperature; TN: thermoneutral; BAT: brown adipose tissue; DCs: dendritic cells; NAFLD: non-alcoholic fatty liver diseases; HFD: high-fat diet; LDL: low-density lipoprotein.

**Table 1 cells-12-00881-t001:** Summary of the effects of housing temperature on the phenotypes observed in mice disease models.

Disease Model	RT (20–24 °C)	TN (28–31 °C)	Ref
Obesity and insulin resistance	Adipose tissue inflammation ↑ Body weight ↑ Insulin resistance ↑ Serum cholesterol and TG ↑	Adipose tissue inflammation ↑↑ Body weight ↑ Insulin resistance ↑ Serum cholesterol and TG ↑↑	[23,82,83]
Atherosclerosis	Atherogenic lesions (*Apoe^−/−^*) ↑ Atherogenic lesions (C57BL/6)- Immune cell infiltration ↑ LDL fraction ↑	Atherogenic lesions (*Apoe^-/-^*) ↑↑ Atherogenic lesions (C57BL/6) ↑ Immune cell infiltration ↑↑ LDL fraction ↑↑	[82,84]
NAFLD	Female resistance disease development	Sex-independence disease development Intestinal permeability ↑ Gram-negative bacteria ↑	[76]
Cancer	Tumor growth and metastasis ↑ CD8+ T cells ↓ Sensitivity of cancer therapies ↓	Tumor growth and metastasis ↓ CD8+ T cells ↑ Sensitivity of cancer therapies ↑	[77,85,86,87]

RT: room temperature; TN: thermoneutral; TG: triglycerides; LDL: low-density lipoprotein; NAFLD: non-alcoholic fatty liver disease.

## Data Availability

Not applicable.

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
