# Peer review of "The Influence of Ambient Temperature on Adipose Tissue Homeostasis, Metabolic Diseases and Cancers"

_cells, 2023, doi:10.3390/cells12060881_

Round 1
Reviewer 1 Report
The review aims to discuss the importance of the temperature of mice housing for human translational studies.
General comments:
-There are already several recent reviews on the same subject: it is not easy to detect what this review brings in addition. Authors shoud emphasized this point in the writing
-the title does not reflect the entire review
-there is a lot of repetitions in the paper between the different paragraphes
- the review is hard to read: the authors mainly described the differences between mice housed at room temperature and mice housed at their thermoneutrality. Most of the time, it is difficult to understand why the authors concluded that the model of mice at thermoneutrality is better for human translational studies.
- the section adipose tissue and systemic inflammation is obsolete. No one talks anymore of M1-activated macrophage or M2 macrophages in the adipose tissues
-it will be good for the readers that the authors clearly specified when it is mild-cold and when it is cold exposure.
Reviewer 2 Report
This review i overall well written.
1. However, it may address some relevant points concerning translational impact of studies using mouse "BAT". First, it may be clarified that interscapular brown adipose tissue (BAT) is one and unique form of thermogenic adipose tissue, and not the sole form of it. Actually, large mammals - such as human - do not have such interscapular BAT. The human thermogenic fat depots are mostly visceral and perivascular, which have little equivalents in mouse. The only exception is the early postnatal life, when there are thermogenic fat cells throughout the subcutanous fat depots in both mouse and human. (Importantly, NOT only in the interscapular region.)
2. Transcriptional landscaping suggests that a young mouse has beige fat depots. And this appears to be analogoues to the human thermogenic fat. Since there is no homeothermy after birth, but the newborn mouse is at a thermoneutral condistion (due to the warmth of ths mother and the syblings in the nest). Hence I feel important to note that this article is slely about the study of thermogenic fat in adult mammals.
3. To this end another limitation arises that should be discussed: an adult human has very small amount of thermogenic fat, which is confined to the perirenal and perivascular regions. This is a limitation in the translation of mouse studies to the human situation. What do we know about the homlogies of the perirenal and periaortic thermogeni fat cells of mouse and human?
4. Beta adrenergic signaling is mentioned as the trigger of "brown" fat development. However, there are other signals which are independent from cold stress, such as interleukin-6, PPARg ligands (such as PAF-metabolites) and mitochondrial RNA, and also the energy oversupply from HFD. It deserves mention that such signals are independent of the Th1 and Th2 cytokine environment.
5. BAT is not infiltrated by inflammatory cells in obesity, unlike the white fat depots. This challenges the concet shown that the Th1/Th2 cytokine environment is decisive for the development of the thermogenic potential in fat. Moreover, it has been challenged that Th2 cytokines (and M2 macrophages) would be involved in thermogenic induction at all. Please address these limitations.
Author Response
Please see the attachement

Reviewer 3 Report
Article: Thermoneutrality: influence of ambient temperature on adipose tissue homeostasis and systemic metabolism
This review article proposed by Rehna Paula Ginting, Ji-Min Lee, and Min-Woo Lee highlights the role of thermoneutrality in systemic metabolism and specifically in adipose tissue. The review topic is extremely important in the field of BAT physiology in mice housed at TN. The writing of the article is clear, objective, instructive, and articulate. The proposed Figures are completely relevant. I really enjoyed reading the manuscript. I have some minor points to contribute to the review.
Minor points:
The abbreviation section was missed but it is necessary. Please, include it.
A final summarizing Figure showing all the review's main points would be nice.
Line 53-57: It is not clear in the manuscript which is the thermoneutral temperature for humans. Please, clarify it.
Lines 57-59: The authors need to indicate the period of time evidenced for significant weight loss by temperature in the cited study. In another study (https://doi.org/10.1038/s42003-022-03895-8), it was shown that housing at different temperatures for up to 33 days did not lead to group differences in body weight, fat-free mass, and fat mass.
Line 91: Please, clarify why NST is preferred to ST during cold exposure.
Author Response
Please see the attachement

Reviewer 4 Report
General comment
. The authors principally start each subsection with an introduction to that particular phenomenon – and then address the thermoneutrality issue for that phenomenon. These introductory sections generally feel too long and tend to dilute the main message of the paper – and sometimes become somewhat controversial or “strange”. I suggest to shorten the introductory parts substantially.
Thus, as an example, the sentence lines 45-46 is not comprehensible. I think the authors should consider starting the review at line 49…
L 50: is this “hot topic” an intended joke? If not, reformulate.
L 51: human ambient temperature sounds strange
L 55: why just obese conditions? (and, finicking, “obese conditions” can’t “live”.
L 63-64: I don't understand why it is elusive?
L 103: this should be a new paragraph.
Figure 1. The authors should extend the line from the green into the yellow to point to the fact that the line extrapolates to the defended body temperature. It seems here that it will extrapolate to about 37 °C (as it should) – but this is not drawn. Importantly, the authors draw the red as if the metabolism increased linearly with temperature above the TNZ. There is neither theoretical nor experimental evidence for this; most observations would indicate an exponential effect of increasing temperature. Also the implication that this heat-induced metabolism can reach the same level as the cold-induced (at 0 °C) is incorrect; mice do not tolerate much increase in environmental temperature above thermoneutrality.
L 118: I have not checked the paper but I would imagine that the nude mice were not examined at their TNZ – which may be slightly higher than that of normal mice. At thermoneutrality, I would assume that the nude mice have a metabolic rate similar to that of normal, and the 80 % is when they are examined at RT.
L 149-188: The authors may discuss the terminology here. They use the expression of BAT (and WAT) whitening for the difference between RT and TN. However, if the authors – as they imply – consider TN the better temperature for translational studies, this is not really a whitening process. Rather, it is a browning induced by the cold of RT. Thus, it is not whitening that should be explained - but browning…
L 191: “previously”? do the authors just mean “used”? “previously” indicates that Jong no longer uses this term…
L 199: I don't understand what the authors here mean. In what way contradicts this what?
L 230-236: without looking up the references, I have difficulties in understanding this. Do all these comments refer to human studies? That should be clearly indicated.
L 279: I cannot see such a limitation if the mouse experiments are performed at the correct temperature.
L 319: I think that much higher contributions than 1 % have been reported. For instance, in the PET studies, it is clear that a large fraction of glucose is taken up by the BAT.
L 328: change to “stated that normal C57BL/6J”
L332: these authors gave lean mass as a percentage. There was no real difference in lean mass – but if fat mass increases in absolute terms, there will be an apparent decrease in lean mass as a percentage.
L 352: are the results contradictory? Or do the authors just mean that the outcome is different when examined at RT and TN? Then it is not contradictory.
L 411: I am unsure what the authors mean. As written, it seems to me that normal Bl6 mice at TN as as good models as are Apoe- at RT. And then principally the normal mice at TN would be better, wouldn't they?
Author Response
Please see the attachement

Round 2
Reviewer 1 Report
The authors change the title for an other formulation than the previous title: Cancer must be added in the title
I appreciated changes that have been made and that improve the review. The revised manuscript remains hard to read. There are still a lot of repetitions in the paper between the different paragraphes and sometimes tue paragraphs are out of place see below for examples
"However, chronic RT housing abolishes the body weight difference found in mice housed at RT and TN [17]": The explanation is on line 133!!!
"The gradual decrease in mice housing temperature from 30°C (TN) to ~20°C - 16°C (mild cold) and ~5°C (severe cold) leads to a gradual increase in mice oxygen consumption [29,30]. This expériment (from 30 to 20°C) was also performed in ref 17 (lane 75)
"Studies have compared the physiological and metabolic differences of mice housed at RT and near their TN zone. Mice housed at RT have higher metabolic rates and consume more food to compensate for their energy deficiency. They prefer to metabolize lipids than carbohydrates and experience sleep deprivation [33,34]". This is also described in reference 17 (lane 74). It is always the same concept that is described from the line 74 to 124, but in different paragraphs.
The review would be more comprehensive after a restructuration of the data of the litterature
"Using nude mice on a C57BL/6 background, Stemmer et al. demonstrated that nude mice had higher EE than wild-type mice when housed at 23°C, whereas 33°C housing abolished the EE difference [38] (lane 137_138). Also it is an important point, without an explanation the readers will not see the importance in the context of thermogenesis.
Authors added a paragraph on T cell subtypes in adipose tissues, the effect of cytokines and of a high fat diet on these populations start at lane 238. This is in part redundant with line 257 "In addition to macrophages, CD4+ and CD8+ T-cells populations are increased in obese conditions. TH2 T-cells are the major T-cell population in lean adipose tissue; however, weight gain shifts the T-cell population into TH1 and cytotoxic T cells and decreases regulatory T cells (Tregs) [72-74]. Dendritic cells (DCs) can also be found in adipose tissue and presumably as an independent contributor to adipose tissue inflammation ".
lane 227 to 272 described AT inflammation, systemic inflammation in obesity and not at thermoneutrality: it is very long and still without important recent discoveries in the field.
I continue to think that the recent discoveries on AT immune cells (and also the compexity in adipocyte subtypes) are important to mention even in a synthetic way even if these different cell types have not yet been studies at thermoneutrality. it will be an added value for the review
I do not understand why the paragraph starting à line 524 is in the paragraph " concluding remarks". I think it should be in the paragraph "inflammation"
In conclusion, the structure of the review deserves to be reconsidered to make the reader fully understand the message of the review on the rodent model's housing to be used when studying metabalic deseases or cancers
